# Sparse Linear Networks with a Fixed Butterfly Structure: Theory and Practice

## Abstract

A butterfly network consists of logarithmically many layers, each with a linear number of non-zero weights (pre-specified). The fast Johnson-Lindenstrauss transform (FJLT) can be represented as a butterfly network followed by a projection onto a random subset of the coordinates. Moreover, a random matrix based on FJLT with high probability approximates the action of any matrix on a vector. Motivated by these facts, we propose to replace a dense linear layer in any neural network by an architecture based on the butterfly network. The proposed architecture significantly improves upon the quadratic number of weights required in a standard dense layer to nearly linear with little compromise in expressibility of the resulting operator. In a collection of wide variety of experiments, including supervised prediction on both the NLP and vision data, we show that this not only produces results that match and often outperform existing well-known architectures, but it also offers faster training and prediction in deployment. To understand the optimization problems posed by neural networks with a butterfly network, we study the optimization landscape of the encoder-decoder network, where the encoder is replaced by a butterfly network followed by a dense linear layer in smaller dimension. Theoretical result presented in the paper explain why the training speed and outcome are not compromised by our proposed approach. Empirically we demonstrate that the network performs as well as the encoder-decoder network.

## 1 Introduction

A butterfly network (see Figure 6 in Appendix A) is a layered graph connecting a layer of $n$ inputs to a layer of $n$ outputs with $O(\log n)$ layers, where each layer contains $2n$ edges. The edges connecting adjacent layers are organized in disjoint gadgets, each gadget connecting a pair of nodes in one layer with a corresponding pair in the next layer by a complete graph. The distance between pairs doubles from layer to layer. This network structure represents the execution graph of the Fast Fourier Transform (FFT) (Cooley and Tukey, 1965), Walsh-Hadamard transform, and many important transforms in signal processing that are known to have fast algorithms to compute matrix-vector products.

Ailon and Chazelle (2009) showed how to use the Fourier (or Hadamard) transform to perform fast Euclidean dimensionality reduction with Johnson and Lindenstrauss (1984) guarantees. The resulting transformation, called Fast Johnson Lindenstrauss Transform (FJLT), was improved in subsequent works (Ailon and Liberty, 2009; Krahmer and Ward, 2011). The common theme in this line of work is to define a fast randomized linear transformation that is composed of a random diagonal matrix, followed by a dense orthogonal transformation which can be represented via a butterfly network, followed by a random projection onto a subset of the coordinates (this research is still active, see e.g. Jain et al. (2020)). In particular, an FJLT matrix can be represented (explicitly) by a butterfly network followed by projection onto a random subset of coordinates (a truncation operator). We refer to such a representation as a truncated butterfly network (see Section 4).

Simple Johnson-Lindenstrauss like arguments show that with high probability for any $W \in \mathbb{R}^{n_2 \times n_1}$ and any $\mathbf{x} \in \mathbb{R}^{n_1}$, $W\mathbf{x}$ is close to $(J_2^T J_2) W (J_1^T J_1) \mathbf{x}$ where $J_1 \in \mathbb{R}^{k_1 \times n_1}$ and $J_2 \in \mathbb{R}^{k_2 \times n_2}$ are both FJLT, and $k_1 = \log n_1, k_2 = \log n_2$ (see Section 4.2 for details). Motivated by this, we propose to replace a dense (fully-connected) linear layer of size $n_2 \times n_1$ in any neural network by the following architecture: $J_1^T W' J_2$, where $J_1, J_2$ can be represented by a truncated butterfly

network and $W'$ is a $k_2 \times k_1$ dense linear layer. The clear advantages of such a strategy are: (1) almost all choices of the weights from a specific distribution, namely the one mimicking FJLT, preserve accuracy while reducing the number of parameters, and (2) the number of weights is nearly linear in the layer width of $W$ (the original matrix). Our empirical results demonstrate that this offers faster training and prediction in deployment while producing results that match and often outperform existing known architectures. Compressing neural networks by replacing linear layers with structured linear transforms that are expressed by fewer parameters have been studied extensively in the recent past. We compare our approach with these related works in Section 3.

Since the butterfly structure adds logarithmic depth to the architecture, it might pose optimization related issues. Moreover, the sparse structure of the matrices connecting the layers in a butterfly network defies the general theoretical analysis of convergence of deep linear networks. We take a small step towards understanding these issues by studying the optimization landscape of a encoder-decoder network (two layer linear neural network), where the encoder layer is replaced by a truncated butterfly network followed by a dense linear layer in fewer parameters. This replacement is motivated by the result of Sarlós (2006), related to fast randomized low-rank approximation of matrices using FJLT (see Section 4.2 for details). We consider this replacement instead of the architecture consisting of two butterfly networks and a dense linear layer as proposed earlier, because it is easier to analyze theoretically. We also empirically demonstrate that our new network with *fewer parameters* performs as well as an encoder-decoder network.

The encoder-decoder network computes the best low-rank approximation of the input matrix. It is well-known that with high probability *a close to optimal* low-rank approximation of a matrix is obtained by either pre-processing the matrix with an FJLT (Sarlós, 2006) or a random sparse matrix structured as given in Clarkson and Woodruff (2009) and then computing the best low-rank approximation from the rows of the resulting matrix[1]. A recent work by Indyk et al. (2019) studies this problem in the supervised setting, where they find the best pre-processing matrix structured as given in Clarkson and Woodruff (2009) from a sample of matrices (instead of using a random sparse matrix). Since an FJLT can be represented by a truncated butterfly network, we emulate the setting of Indyk et al. (2019) but learn the pre-processing matrix structured as a truncated butterfly network.

## 2 Our Contribution and Potential Impact

We provide an empirical report, together with a theoretical analysis to justify our main idea of using sparse linear layers with a fixed butterfly network in deep learning. Our findings indicate that this approach, which is well rooted in the theory of matrix approximation and optimization, can offer significant speedup and energy saving in deep learning applications. Additionally, we believe that this work would encourage more experiments and theoretical analysis to better understand the optimization and generalization of our proposed architecture (see Future Work section).

**On the empirical side** – The outcomes of the following experiments are reported:

(1) In Section 6.1, we replace a dense linear layer in the standard state-of-the-art networks, for both image and language data, with an architecture that constitutes the composition of (a) truncated butterfly network, (b) dense linear layer in smaller dimension, and (c) transposed truncated butterfly network (see Section 4.2). The structure parameters are chosen so as to keep the number of weights near linear (instead of quadratic).

(2) In Sections 6.2 and 6.3, we train a linear encoder-decoder network in which the encoder is replaced by a truncated butterfly network followed by a dense linear layer in smaller dimension. These experiments support our theoretical result. The network structure parameters are chosen so as to keep the number of weights in the (replaced) encoder near linear in the input dimension. Our results (also theoretically) demonstrate that this has little to no effect on the performance compared to the standard encoder-decoder network.

(3) In Section 7, we learn the best pre-processing matrix structured as a truncated butterfly network to perform low-rank matrix approximation from a given sample of matrices. We compare our results

---

[1]The pre-processing matrix is multiplied from the left.

to that of Indyk et al. (2019), which learn the pre-processing matrix structured as given in Clarkson and Woodruff (2009).

**On the theoretical side** – The optimization landscape of linear neural networks with dense matrices have been studied by Baldi and Hornik (1989), and Kawaguchi (2016). The theoretical part of this work studies the optimization landscape of the linear encoder-decoder network in which the encoder is replaced by a truncated butterfly network followed by a dense linear layer in smaller dimension. We call such a network as the encoder-decoder butterfly network. We give an overview of our main result, Theorem 1, here. Let $X \in \mathbb{R}^{n \times d}$ and $Y \in \mathbb{R}^{m \times d}$ be the data and output matrices respectively. Then the *encoder-decoder butterfly network* is given as $\overline{Y} = DEBX$, where $D \in \mathbb{R}^{m \times k}$ and $E \in \mathbb{R}^{k \times \ell}$ are dense layers, $B$ is an $\ell \times n$ truncated butterfly network (product of $\log n$ sparse matrices) and $k \leq \ell \leq m \leq n$ (see Section 5). The objective is to learn $D, E$ and $B$ that minimizes $||\overline{Y} - Y||_{\text{F}}^2$. Theorem 1 shows how the loss at the critical points of such a network depends on the eigenvalues of the matrix $\Sigma = YX^T B^T (BXX^T B^T)^{-1} BXY^T$ [2]. In comparison, the loss at the critical points of the encoder-decoder network (without the butterfly network) depends on the eigenvalues of the matrix $\Sigma' = YX^T (XX^T)^{-1} XY^T$ (Baldi and Hornik, 1989). In particular, the loss depends on how the learned matrix $B$ changes the eigenvalues of $\Sigma'$. If we learn only for an optimal $D$ and $E$, keeping $B$ fixed (as done in the experiment in Section 6.3) then it follows from Theorem 1 that every local minima is a global minima and that the loss at the local/global minima depends on how $B$ changes the top $k$ eigenvalues of $\Sigma'$. This inference together with a result by Sarlós (2006) is used to give a worst-case guarantee in the special case when $Y = X$ (called auto-encoders that capture PCA; see the below Theorem 1).

## 3    RELATED WORK

Important transforms like discrete Fourier, discrete cosine, Hadamard and many more satisfy a property called *complementary low-rank* property, recently defined by Li et al. (2015). For an $n \times n$ matrix satisfying this property related to approximation of specific sub-matrices by low-rank matrices, Michielssen and Boag (1996) and O'Neil et al. (2010) developed the butterfly algorithm to compute the product of such a matrix with a vector in $O(n \log n)$ time. The butterfly algorithm factorizes such a matrix into $O(\log n)$ many matrices, each with $O(n)$ sparsity. In general, the butterfly algorithm has a pre-computation stage which requires $O(n^2)$ time (O'Neil et al., 2010; Seljebotn, 2012). With the objective of reducing the pre-computation cost Li et al. (2015); Li and Yang (2017) compute the butterfly factorization for an $n \times n$ matrix satisfying the complementary low-rank property in $O(n^{\frac{3}{2}})$ time. This line of work does not learn butterfly representations for matrices or apply it in neural networks, and is incomparable to our work.

A few works in the past have used deep learning models with structured matrices (as hidden layers). Such structured matrices can be described using fewer parameters compared to a dense matrix, and hence a representation can be learned by optimizing over a fewer number of parameters. Examples of structured matrices used include low-rank matrices (Denil et al., 2013; Sainath et al., 2013), circulant matrices (Cheng et al., 2015; Ding et al., 2017), low-distortion projections (Yang et al., 2015), Toeplitz like matrices (Sindhwani et al., 2015; Lu et al., 2016; Ye et al., 2018), Fourier-related transforms (Moczulski et al., 2016) and matrices with low-displacement rank (Thomas et al., 2018). Recently Alizadeh et al. (2020) demonstrated the benefits of replacing the pointwise convolutional layer in CNN's by a butterfly network. Other works by Mocanu et al. (2018); Lee et al. (2019); Wang et al. (2020); Verdenius et al. (2020) consider a different approach to sparsify neural networks. The works closest to ours are by Yang et al. (2015), Moczulski et al. (2016), and Dao et al. (2020) and we make a comparison below.

Yang et al. (2015) and Moczulski et al. (2016) attempt to replace dense linear layers with a stack of structured matrices, including a butterfly structure (the Hadamard or the Cosine transform), but they do not place trainable weights on the edges of the butterfly structure as we do. Note that adding these trainable weights does not compromise the run time benefits in prediction, while adding to the expressiveness of the network in our case. Dao et al. (2020) replace handcrafted structured sub-networks in machine learning models by a *kaleidoscope* layer, which consists of compositions of butterfly matrices. This is motivated by the fact that the kaleidoscope hierarchy captures a structured matrix exactly and optimally in terms of multiplication operations required to perform the matrix

---

[2]At a critical point the gradient of the loss function with respect to the parameters in the network is zero.

vector product operation. Their work differs from us as we propose to replace any dense linear layer in a neural network (instead of a structured sub-network) by the architecture proposed in Section 4.2. Our approach is motivated by theoretical results which establish that this can be done with almost no loss in representation.

Finally, Dao et al. (2019) show that butterfly representations of standard transformations like discrete Fourier, discrete cosine, Hadamard mentioned above can be learnt efficiently. They additionally show the following: a) for the benchmark task of compressing a single hidden layer model they compare the network constituting of a composition of butterfly networks with the classification accuracy of a fully-connected linear layer and b) in ResNet a butterfly sub-network is added to get an improved result. In comparison, our approach to replace a dense linear layer by the proposed architecture in Section 4.2 is motivated by well-known theoretical results as mentioned previously, and the results of the comprehensive list of experiments in Section 6.1 support our proposed method.

## 4 PROPOSED REPLACEMENT FOR A DENSE LINEAR LAYER

In Section 4.1, we define a truncated butterfly network, and in Section 4.2 we motivate and state our proposed architecture based on truncated butterfly network to replace a dense linear layer in any neural network. All logarithms are in base 2, and $[n]$ denotes the set $\{1, \ldots, n\}$.

### 4.1 TRUNCATED BUTTERFLY NETWORK

**Definition 4.1** (Butterfly Network). *Let $n$ be an integral power of 2. Then an $n \times n$ butterfly network $B$ (see Figure 6) is a stack of of $\log n$ linear layers, where in each layer $i \in \{0, \ldots, \log n - 1\}$, a bipartite clique connects between pairs of nodes $j_1, j_2 \in [n]$, for which the binary representation of $j_1 - 1$ and $j_2 - 1$ differs only in the $i$'th bit. In particular, the number of edges in each layer is $2n$.*

In what follows, a *truncated butterfly network* is a butterfly network in which the deepest layer is truncated, namely, only a subset of $\ell$ neurons are kept and the remaining $n - \ell$ are discarded. The integer $\ell$ is a tunable parameter, and the choice of neurons is always assumed to be sampled uniformly at random and fixed throughout training in what follows. The effective number of parameters (trainable weights) in a truncated butterfly network is at most $2n \log \ell + 6n$, for any $\ell$ and any choice of neurons selected from the last layer.[3] We include a proof of this simple upper bound in Appendix F for lack of space (also, refer to Ailon and Liberty (2009) for a similar result related to computation time of truncated FFT). The reason for studying a truncated butterfly network follows (for example) from the works (Ailon and Chazelle, 2009; Ailon and Liberty, 2009; Krahmer and Ward, 2011). These papers define randomized linear transformations with the Johnson-Lindenstrauss property and an efficient computational graph which essentially defines the truncated butterfly network. In what follows, we will collectively denote these constructions by FJLT. [4]

### 4.2 MATRIX APPROXIMATION USING BUTTERFLY NETWORKS

We begin with the following proposition, following known results on matrix approximation (proof in Appendix B).

**Proposition 1.** *Suppose $J_1 \in \mathbb{R}^{k_1 \times n_1}$ and $J_2 \in \mathbb{R}^{k_2 \times n_2}$ are matrices sampled from FJLT distribution, and let $W \in \mathbb{R}^{n_2 \times n_1}$. Then for the random matrix $W' = (J_2^T J_2) W (J_1^T J_1)$, any unit vector $\mathbf{x} \in \mathbb{R}^{n_1}$ and any $\epsilon \in (0, 1)$, $\Pr\left[ \|W'\mathbf{x} - W\mathbf{x}\| \leq \epsilon \|W\| \right] \geq 1 - e^{-\Omega(\min\{k_1, k_2\}\epsilon^2)}$.*

From Proposition 1 it follows that $W'$ approximates the action of $W$ with high probability on any given input vector. Now observe that $W'$ is equal to $J_2^T \tilde{W} J_1$, where $\tilde{W} = J_2 W J_1^T$. Since $J_1$ and $J_2$ are FJLT, they can be represented by a truncated butterfly network, and hence it is conceivable to replace a dense linear layer connecting $n_1$ neurons to $n_2$ neurons (containing $n_1 n_2$ variables) in any

---

[3]Note that if $n$ is not a power of 2 then we work with the first $n$ columns of the $\ell \times n'$ truncated butterfly network, where $n'$ is the closest number to $n$ that is greater than $n$ and is a power of 2.

[4]To be precise, the construction in Ailon and Chazelle (2009), Ailon and Liberty (2009), and Krahmer and Ward (2011) also uses a random diagonal matrix, but the values of the diagonal entries can be 'absorbed' inside the weights of the first layer of the butterfly network.

neural network with a composition of three gadgets: a truncated butterfly network of size $k_1 \times n_1$, followed by a dense linear layer of size $k_2 \times k_1$, followed by the transpose of a truncated butterfly network of size $k_2 \times n_2$. In Section 6.1, we replace dense linear layers in common deep learning networks with our proposed architecture, where we set $k_1 = \log n_1$ and $k_2 = \log n_2$.

## 5 ENCODER-DECODER BUTTERFLY NETWORK

Let $X \in \mathbb{R}^{n \times d}$, and $Y \in \mathbb{R}^{m \times d}$ be data and output matrices respectively, and $k \leq m \leq n$. Then the encoder-decoder network for $X$ is given as

$$\overline{Y} = DEX$$

where $E \in \mathbb{R}^{k \times n}$, and $D \in \mathbb{R}^{m \times k}$ are called the encoder and decoder matrices respectively. For the special case when $Y = X$, it is called auto-encoders. The optimization problem is to learn matrices $D$ and $E$ such that $||Y - \overline{Y}||_F^2$ is minimized. The optimal solution is denoted as $Y^*, D^*$ and $E^{*5}$. In the case of auto-encoders $X^* = X_k$, where $X_k$ is the best rank $k$ approximation of $X$. In this section, we study the optimization landscape of the encoder-decoder butterfly network : an encoder-decoder network, where the encoder is replaced by a truncated butterfly network followed by a dense linear layer in smaller dimension. Such a replacement is motivated by the following result from Sarlós (2006), in which $\Delta_k = ||X_k - X||_F^2$.

**Proposition 2.** *Let $X \in \mathbb{R}^{n \times d}$. Then with probability at least $1/2$, the best rank $k$ approximation of $X$ from the rows of $JX$ (denoted $J_k(X)$), where $J$ is sampled from an $\ell \times n$ FJLT distribution and $\ell = (k \log k + k/\epsilon)$ satisfies $||J_k(X) - X||_F^2 \leq (1+\epsilon)\Delta_k$.*

Proposition 2 suggests that in the case of auto-encoders we could replace the encoder with a truncated butterfly network of size $\ell \times n$ followed by a dense linear layer of size $k \times \ell$, and obtain a network with fewer parameters but loose very little in terms of representation. Hence, it is worthwhile investigating the representational power of the encoder-decoder butterfly network

$$\overline{Y} = DEBX . \tag{1}$$

Here, $X$, $Y$ and $D$ are as in the encoder-decoder network, $E \in \mathbb{R}^{k \times \ell}$ is a dense matrix, and $B$ is an $\ell \times n$ truncated butterfly network. In the encoder-decoder butterfly network the encoding is done using $EB$, and decoding is done using $D$. This reduces the number of parameters in the encoding matrix from $kn$ (as in the encoder-decoder network) to $k\ell + O(n \log \ell)$. Again the objective is to learn matrices $D$ and $E$, and the truncated butterfly network $B$ such that $||Y - \overline{Y}||_F^2$ is minimized. The optimal solution is denoted as $Y^*, D^*, E^*$, and $B^*$. Theorem 1 shows that the loss at a critical point of such a network depends on the eigenvalues of $\Sigma(B) = YX^TB^T(BXX^TB^T)^{-1}XY^T$, when $BXX^TB^T$ is invertible and $\Sigma(B)$ has $\ell$ distinct positive eigenvalues. The loss $\mathcal{L}$ is defined as $||\overline{Y} - Y||_F^2$.

**Theorem 1.** *Let $D, E$ and $B$ be a point of the encoder-decoder network with a truncated butterfly network satisfying the following: a) $BXX^TB^T$ is invertible, b) $\Sigma(B)$ has $\ell$ distinct positive eigenvalues $\lambda_1 > \ldots > \lambda_\ell$, and c) the gradient of $\mathcal{L}(\overline{Y})$ with respect to the parameters in $D$ and $E$ matrix is zero. Then corresponding to this point (and hence corresponding to every critical point) there is an $I \subseteq [\ell]$ such that $\mathcal{L}(\overline{Y})$ at this point is equal to $tr(YY^T) - \sum_{i \in I} \lambda_i$. Moreover if the point is a local minima then $I = [k]$.*

The proof of Theorem 1 is given in Appendix C. We also compare our result with that of Baldi and Hornik (1989) and Kawaguchi (2016), which study the optimization landscape of dense linear neural networks in Appendix C. From Theorem 1 it follows that if $B$ is fixed and only $D$ and $E$ are trained then a local minima is indeed a global minima. We use this to claim a worst-case guarantee using a two-phase learning approach to train an auto-encoder. In this case the optimal solution is denoted as $B_k(Y), D_B$, and $E_B$. Observe that when $Y = X$, $B_k(X)$ is the best rank $k$ approximation of $X$ computed from the rows of $BX$.

**Two phase learning for auto-encoder**: Let $\ell = k \log k + k/\epsilon$ and consider a two phase learning strategy for auto-encoders, as follows: In phase one $B$ is sampled from an FJLT distribution, and then only $D$ and $E$ are trained keeping $B$ fixed. Suppose the algorithm learns $D'$ and $E'$ at the end

---

[5]Possibly multiple $D^*$ and $E^*$ exist such that $Y^* = D^*E^*X$.

| Dataset Name | Task | Model |
|---|---|---|
| Cifar-10 Krizhevsky (2012) | Image classification | EfficientNet Tan and Le (2019) |
| Cifar-10 Krizhevsky (2012) | Image classification | PreActResNet18 He et al. (2016) |
| Cifar-100 Krizhevsky (2012) | Image classification | seresnet152 Hu et al. (2020) |
| Imagenet Deng et al. (2009) | Image classification | senet154 Hu et al. (2020) |
| CoNLL-03 Tjong Kim Sang and De Meulder (2003) | Named Entity Recognition (English) | Flair's Sequence Tagger Akbik et al. (2018) Akbik et al. (2019) |
| CoNLL-03 Tjong Kim Sang and De Meulder (2003) | Named Entity Recognition (German) | Flair's Sequence Tagger Akbik et al. (2018) Akbik et al. (2019) |
| Penn Treebank (English) Marcus et al. (1993) | Part-of-Speech Tagging | Flair's Sequence Tagger Akbik et al. (2018) Akbik et al. (2019) |

Table 1: Data and the corresponding architectures used in the fast matrix multiplication using butterfly matrices experiments.

of phase one, and $X' = D'E'B$. Then Theorem 1 guarantees that, assuming $\Sigma(B)$ has $\ell$ distinct positive eigenvalues and $D', E'$ are a local minima, $D' = D_B$, $E' = E_B$, and $X' = B_k(X)$. Namely $X'$ is the best rank $k$ approximation of $X$ from the rows of $BX$. From Proposition 2 with probability at least $\frac{1}{2}$, $\mathcal{L}(X') \leq (1 + \epsilon)\Delta_k$. In the second phase all three matrices are trained to improve the loss. In Sections 6.2 and 6.3 we train an encoder-decoder butterfly network using the standard gradient descent method. In these experiments the truncated butterfly network is initialized by sampling it from an FJLT distribution, and $D$ and $E$ are initialized randomly as in Pytorch.

# 6    EXPERIMENTS ON DENSE LAYER REPLACEMENT AND ENCODER-DECODER BUTTERFLY NETWORK

In this section we report the experimental results based on the ideas presented in Sections 4.2 and 5.

## 6.1    REPLACING DENSE LINEAR LAYERS BY THE PROPOSED ARCHITECTURE

This experiment replaces a dense linear layer of size $n_2 \times n_1$ in common deep learning architectures with the network proposed in Section 4.2.[6] The truncated butterfly networks are initialized by sampling it from the FJLT distribution, and the dense matrices are initialized randomly as in Pytorch. We set $k_1 = \log n_1$ and $k_2 = \log n_2$. The datasets and the corresponding architectures considered are summarized in Table 1. For each dataset and model, the objective function is the same as defined in the model, and the generalization and convergence speed between the original model and the modified one (called the butterfly model for convenience) are compared. Figure 7 in Appendix D.1 reports the number of parameters in the dense linear layer of the original model, and in the replaced network, and Figure 8 in Appendix D.1 displays the number of parameter in the original model and the butterfly model. In particular, Figure 7 shows the significant reduction in the number of parameters obtained by the proposed replacement. On the left of Figure 1, the test accuracy of the original model and the butterfly model is reported, where the black vertical lines denote the error bars corresponding to standard deviation, and the values above the rectangles denote the average accuracy. On the right of Figure 1 observe that the test accuracy for the butterfly model trained with stochastic gradient descent is even better than the original model trained with Adam in the first few epochs. Figure 12 in Appendix D.1 compares the test accuracy in the the first 20 epochs of the original and butterfly model. The results for the NLP tasks in the interest of space are reported in Figure 9, Appendix D.1. The training and inference times required for the original model and the butterfly model in each of these experiments are reported in Figures 10 and 11 in Appendix D.1. We remark that the modified architecture is also trained for fewer epochs. In almost all the cases the modified architecture does better than the normal architecture, both in the rate of convergence and in the final accuracy/$F1$ score. Moreover, the training time for the modified architecture is less.

## 6.2    ENCODER-DECODER BUTTERFLY NETWORK WITH SYNTHETIC GAUSSIAN AND REAL DATA

This experiment tests whether gradient descent based techniques can be used to train encoder-decoder butterfly network. In all the experiments in this section $Y = X$. Five types of data matrices are tested, whose attributes are specified in Table 2.[7] Two among them are random and

---

[6]In all the architectures considered the final linear layer before the output layer is replaced, and $n_1$ and $n_2$ depend on the architecture.

[7]In Table 2 HS-SOD denotes a dataset for hyperspectral images from natural scenes (Imamoglu et al., 2018).

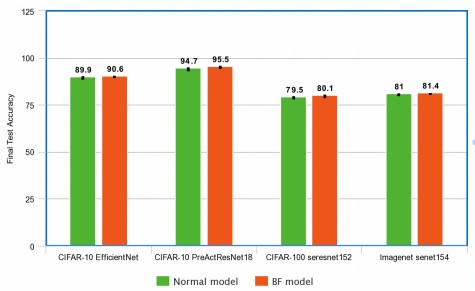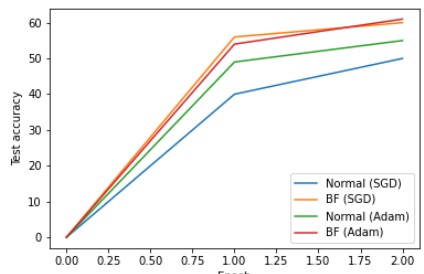

Figure 1: Left: comparison of final test accuracy with different image classification models and data sets; Right: comparison of test accuracy in the first few epochs with different models and optimizers on CIFAR-10 with PreActResNet18

three are constructed using standard public real image datasets. In the interest of space, the construction of the data matrices is explained in Appendix D.2. For the matrices constructed from the image datasets, the input coordinates are randomly permuted, which ensures the network cannot take advantage of the spatial structure in the data. For each of the data matrices the loss obtained via training the truncated butterfly network with the Adam optimizer is compared to $\Delta_k$ (denoted as PCA) and $||J_k(X) - X||_{\mathrm{F}}^2$ where $J$ is an $\ell \times n$ matrix sampled from the FJLT distribution (denoted as FJLT+PCA). Figure 2 reports the loss on Gaussian 1 and MNIST, whereas Figure 13 in Appendix D.2 reports the loss for the remaining data matrices. Observe that for all values of $k$ the loss for the encoder-decoder butterfly network is almost equal to $\Delta_k$, and is in fact $\Delta_k$ for small and large values of $k$.

| Name | $n$ | $d$ | rank |
|---|---|---|---|
| Gaussian 1 | 1024 | 1024 | 32 |
| Gaussian 2 | 1024 | 1024 | 64 |
| MNIST | 1024 | 1024 | 1024 |
| Olivetti | 1024 | 4096 | 1024 |
| HS-SOD | 1024 | 768 | 768 |

Table 2: Data used in the truncated butterfly auto-encoder reconstruction experiments

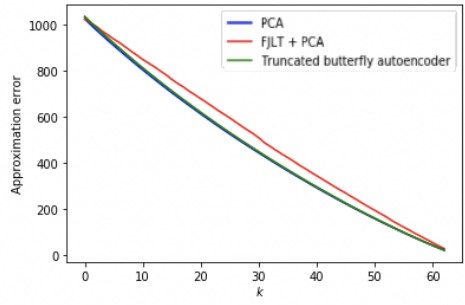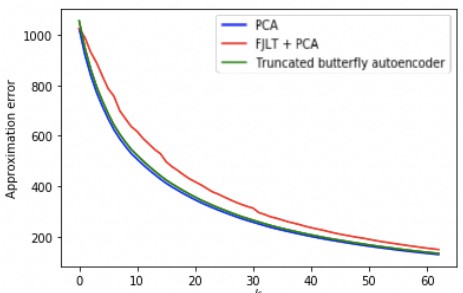

Figure 2: Approximation error on data matrix with various methods for various values of $k$. Left: Gaussian 1 data, Right: MNIST data

## 6.3 TWO-PHASE LEARNING FOR ENCODER-DECODER BUTTERFLY NETWORK

This experiment is similar to the experiment in Section 6.2 but the training in this case is done in two phases. In the first phase, $B$ is fixed and the network is trained to determine an optimal $D$ and $E$. In the second phase, the optimal $D$ and $E$ determined in phase one are used as the initialization, and the

network is trained over $D, E$ and $B$ to minimize the loss. Theorem 1 ensures worst-case guarantees for this two phase training (see below the theorem). Figure 3 reports the approximation error of an image from Imagenet. The red and green lines in Figure 3 correspond to the approximation error at the end of phase one and two respectively.

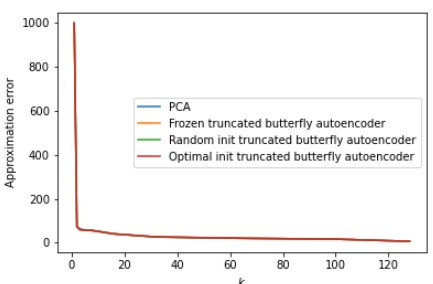 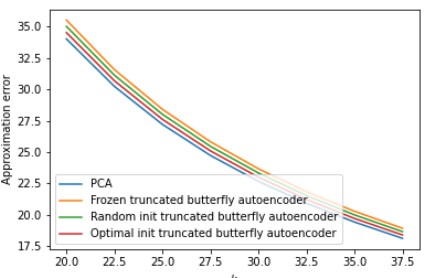

Figure 3: Approximation error achieved by different methods and the same zoomed on in the right

# 7 Sketching Algorithm for Low-Rank Matrix Decomposition Problem Using Butterfly Network

The recent influential work by Indyk et al. (2019) considers a supervised learning approach to compute an $\ell \times n$ pre-conditioning matrix $B$ for low-rank approximation of $n \times d$ matrices. The matrix $B$ has a fixed sparse structure as in Clarkson and Woodruff (2009), each column as one non-zero entry (chosen randomly) which are learned to minimize the loss over a training set of matrices. In this section, we present experiments with the setting being similar to that in Indyk et al. (2019), except that $B$ is now represented as an $\ell \times n$ truncated butterfly network. Our setting is similar to that in Indyk et al. (2019), except that $B$ is now represented as an $\ell \times n$ truncated butterfly network. Our experiments suggests that indeed a learned truncated butterfly network does better than a random matrix, and even a learned $B$ as in Indyk et al. (2019).

**Setup**: Suppose $X_1, \ldots, X_t \in \mathbb{R}^{n \times d}$ are training matrices sampled from a distribution $\mathcal{D}$. Then a $B$ is computed that minimizes the following empirical loss: $\sum_{i \in [t]} ||X_i - B_k(X_i)||_{\mathrm{F}}^2$. We compute $B_k(X_i)$ using truncated SVD of $BX_i$ (as in Algorithm 1, Indyk et al. (2019)). Similar to Indyk et al. (2019), the matrix $B$ is learned by the back-propagation algorithm that uses a differentiable SVD implementation to calculate the gradients, followed by optimization with Adam such that the butterfly structure of $B$ is maintained. The learned $B$ can be used as the pre-processing matrix for any matrix in the future. The test error for a matrix $B$ and a test set $\mathsf{Te}$ is defined as follows:

$$\mathrm{Err}_{\mathsf{Te}}(B) = \mathbf{E}_{X \sim \mathsf{Te}} \left[ ||X - B_k(X)||_{\mathrm{F}}^2 \right] - \mathrm{App}_{\mathsf{Te}}, \quad \text{where } \mathrm{App}_{\mathsf{Te}} = \mathbf{E}_{X \sim \mathsf{Te}} \left[ ||X - X_k||_{\mathrm{F}}^2 \right] .$$

**Experiments and Results**: The experiments are performed on the datasets shown in Table 3. In HS-SOD Imamoglu et al. (2018) and CIFAR-10 Krizhevsky (2012) 400 training matrices ($t = 400$), and 100 test matrices are sampled, while in Tech 200 training matrices ($t = 200$), and 95 test matrices are sampled. In Tech Davido et al. (2004) each matrix has 835,422 rows but on average only 25,389 rows and 195 columns contain non-zero entries. For the same reason as in Section 6.2 in each dataset, the coordinates of each row are randomly permuted. Some of the matrices in the datasets have much larger singular values than the others, and to avoid imbalance in the dataset, the matrices are normalized so that their top singular values are all equal, as done in Indyk et al. (2019). For each of the datasets, the test error for the learned $B$ via our truncated butterfly structure

| Name | $n$ | $d$ |
|---|---|---|
| HS-SOD 1 | 1024 | 768 |
| CIFAR-10 | 32 | 32 |
| Tech | 25,389 | 195 |

Table 3: Data used in the Sketching algorithm for low-rank matrix decomposition experiments.

is compared to the test errors for the following three cases: 1) $B$ is a learned as a sparse sketching matrix as in Indyk et al. (2019), b) $B$ is a random sketching matrix as in Clarkson and Woodruff (2009), and c) $B$ is an $\ell \times n$ Gaussian matrix. Figure 4 compares the test error for $\ell = 20$, and $k = 10$, where $\text{App}_{\text{Te}} = 10.56$. Figure 14 in Appendix E compares the test errors of the different methods in the extreme case when $k = 1$, and Figure 15 in Appendix E compares the test errors of the different methods for various values of $\ell$. Table 4 in Appendix E in Appendix E reports the test error for different values of $\ell$ and $k$. Figure 16 in in Appendix E shows the test error for $\ell = 20$ and $k = 10$ during the training phase on HS-SOD. In Figure 16 it is observed that the butterfly learned is able to surpass sparse learned after a merely few iterations.

Figure 5 compares the test error for the learned $B$ via our truncated butterfly structure to a learned matrix $B$ with $N$ non-zero entries in each column – the $N$ non-zero location for each column are chosen uniformly at random. The reported test errors are on HS-SOD, when $\ell = 20$ and $k = 10$. Interestingly, the error for butterfly learned is not only less than the error for sparse learned ($N = 1$ as in (Indyk et al., 2019)) but also less than than the error for dense learned ($N = 20$). In particular, our results indicate that using a learned butterfly sketch can significantly reduce the approximation loss compared to using a learned sparse sketching matrix.

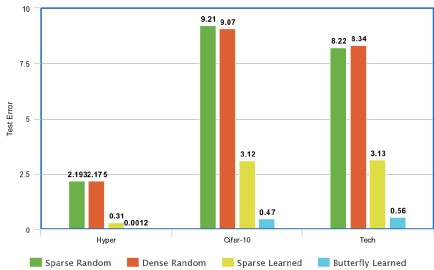

Figure 4: Test error by different sketching matrices on different data sets

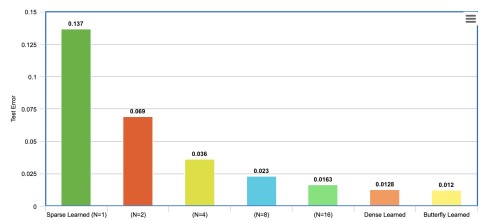

Figure 5: Test errors for various values of $N$ and a learned butterfly matrix

# 8    DISCUSSION AND FUTURE WORK

**Discussion**: Among other things, this work showed that it is beneficial to replace dense linear layer in deep learning architectures with a more compact architecture (in terms of number of parameters), using truncated butterfly networks. This approach is justified using ideas from efficient matrix approximation theory from the last two decades. however, results in additional logarithmic depth to the network. This issue raises the question of whether the extra depth may harm convergence of gradient descent optimization. To start answering this question, we show, both empirically and theoretically, that in linear encoder-decoder networks in which the encoding is done using a butterfly network, this typically does not happen. To further demonstrate the utility of truncated butterfly networks, we consider a supervised learning approach as in Indyk et al. (2019), where we learn how to derive low rank approximations of a distribution of matrices by multiplying a pre-processing linear operator represented as a butterfly network, with weights trained using a sample of the distribution.

**Future Work**: The main open questions arising from the work are related to better understanding the optimization landscape of butterfly networks. The current tools for analysis of deep linear networks do not apply for these structures, and more theory is necessary. It would be interesting to determine whether replacing dense linear layers in any network, with butterfly networks as in Section 4.2 *harms* the convergence of the original matrix. Another direction would be to check empirically whether adding non-linear gates between the layers (logarithmically many) of a butterfly network improves the performance of the network. In the experiments in Section 6.1, we have replaced a single dense layer by our proposed architecture. It would be worthwhile to check whether replacing multiple dense linear layers in the different architectures harms the final accuracy. Similarly, it might be insightful to replace a convolutional layer by an architecture based on truncated butterfly network. Finally, since our proposed replacement reduces the number of parameters in the network, it might be possible to empirically show that the new network is more resilient to over-fitting.

ACKNOWLEDGEMENT

This project has received funding from European Union's Horizon 2020 research and innovation program under grant agreement No 682203 -ERC-[ Inf-Speed-Tradeoff].

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

## A  BUTTERFLY DIAGRAM FROM SECTION 1

Figure 6 referred to in the introduction is given here.

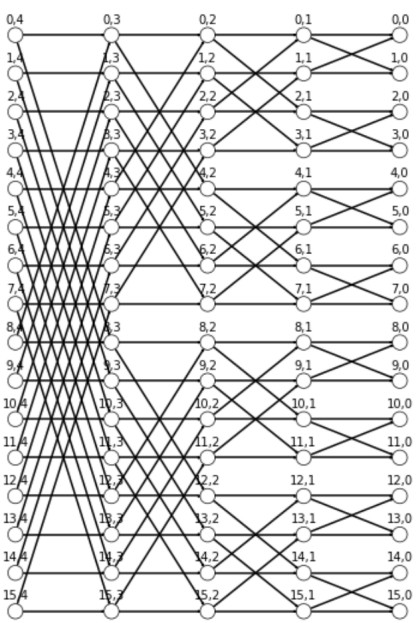
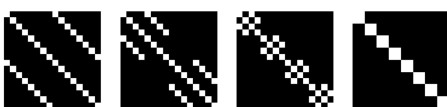

Figure 6: A $16 \times 16$ butterfly network represented as a 4-layered graph on the left, and as product of 4 sparse matrices on the right. The white entries are the non-zero entries of the matrices.

## B  PROOF OF PROPOSITION 1

The proof of the proposition will use the following well known fact (Lemma B.1 below) about FJLT (more generally, JL) distributions (see Ailon and Chazelle (2009); Ailon and Liberty (2009); Krahmer and Ward (2011)).

**Lemma B.1.** *Let* $\mathbf{x} \in \mathbb{R}^n$ *be a unit vector, and let* $J \in \mathbb{R}^{k \times n}$ *be a matrix drawn from an FJLT distribution. Then for all* $\epsilon < 1$ *with probability at least* $1 - e^{-\Omega(k\epsilon^2)}$:

$$\|\mathbf{x} - J^T J \mathbf{x}\| \le \epsilon . \tag{2}$$

By Lemma B.1 we have that with probability at least $1 - e^{-\Omega(k_1 \epsilon^2)}$,

$$\|\mathbf{x} - J_1^T J_1 \mathbf{x}\| \le \epsilon \|\mathbf{x}\| = \epsilon . \tag{3}$$

Henceforth, we condition on the event $\|\mathbf{x} - J_1^T J_1 \mathbf{x}\| \le \epsilon \|\mathbf{x}\|$. Therefore, by the definition of spectral norm $\|W\|$ of $W$:

$$\|W\mathbf{x} - W J_1^T J_1 \mathbf{x}\| \le \epsilon \|W\| . \tag{4}$$

Now apply Lemma B.1 again on the vector $W J_1^T J_1 \mathbf{x}$ and transformation $J_2$ to get that with probability at least $1 - e^{-\Omega(k_2 \epsilon^2)}$,

$$\|W J_1^T J_1 \mathbf{x} - J_2^T J_2 W J_1^T J_1 \mathbf{x}\| \le \epsilon \|W J_1^T J_1 \mathbf{x}\|. \tag{5}$$

Henceforth, we condition on the event $\|W J_1^T J_1 \mathbf{x} - J_2^T J_2 W J_1^T J_1 \mathbf{x}\| \le \epsilon \|W J_1^T J_1 \mathbf{x}\|$. To bound the last right hand side, we use the triangle inequality together with (4):

$$\|W J_1^T J_1 \mathbf{x}\| \le \|W\mathbf{x}\| + \epsilon \|W\| \le \|W\|(1 + \epsilon). \tag{6}$$

Combining (5) and (6) gives:

$$\|W J_1^T J_1 \mathbf{x} - J_2^T J_2 W J_1^T J_1 \mathbf{x}\| \le \epsilon \|W\|(1 + \epsilon). \tag{7}$$

Finally,

$$
\begin{aligned}
\|J_2^T J_2 W J_1^T J_1 \mathbf{x} - W\mathbf{x}\| &= \|(J_2^T J_2 W J_1^T J_1 \mathbf{x} - W J_1^T J_1 \mathbf{x}) + (W J_1^T J_1 \mathbf{x} - W\mathbf{x})\| \\
&\leq \epsilon \|W\|(1+\epsilon) + \epsilon\|W\| \\
&= \|W\|\epsilon(2+\epsilon) \leq 3\|W\|\epsilon \,,
\end{aligned}
\tag{8}
$$

where the first inequality is from the triangle inequality together with (4) and (7), and the second inequality is from the bound on $\epsilon$. The proposition is obtained by adjusting the constants hiding inside the $\Omega()$ notation in the exponent in the proposition statement.

## C  PROOF OF THEOREM 1

We first note that our result continues to hold even if $B$ in the theorem is replaced by any structured matrix. For example the result continues to hold if $B$ is an $\ell \times n$ matrix with one non-zero entry per column, as is the case with a random sparse sketching matrix Clarkson and Woodruff (2009). We also compare our result with that Baldi and Hornik (1989); Kawaguchi (2016).

**Comparison with Baldi and Hornik (1989) and Kawaguchi (2016)**: The critical points of the encoder-decoder network are analyzed in Baldi and Hornik (1989). Suppose the eigenvalues of $YX^T(XX^T)^{-1}XY^T$ are $\gamma_1 > \ldots > \gamma_m > 0$ and $k \leq m \leq n$. Then they show that corresponding to a critical point there is an $I \subseteq [m]$ such that the loss at this critical point is equal to $\text{tr}(YY^T) - \sum_{i \in I} \gamma_i$, and the critical point is a local/global minima if and only if $I = [k]$. Kawaguchi (2016) later generalized this to prove that a local minima is a global minima for an arbitrary number of hidden layers in a linear neural network if $m \leq n$. Note that since $\ell \leq n$ and $m \leq n$ in Theorem 1, replacing $X$ by $BX$ in Baldi and Hornik (1989) or Kawaguchi (2016) does not imply Theorem 1 as it is.

Next, we introduce a few notation before delving into the proof. Let $r = (\overline{Y} - Y)^T$, and $\text{vec}(r) \in \mathbb{R}^{md}$ is the entries of $r$ arranged as a vector in column-first ordering, $(\nabla_{\text{vec}(D^T)}\mathcal{L}(\overline{Y}))^T \in \mathbb{R}^{mk}$ and $(\nabla_{\text{vec}(E^T)}\mathcal{L}(\overline{Y}))^T \in \mathbb{R}^{k\ell}$ denote the partial derivative of $\mathcal{L}(\overline{Y})$ with respect to the parameters in $\text{vec}(D^T)$ and $\text{vec}(E^T)$ respectively. Notice that $\nabla_{\text{vec}(D^T)}\mathcal{L}(\overline{Y})$ and $\nabla_{\text{vec}(E^T)}\mathcal{L}(\overline{Y})$ are row vectors of size $mk$ and $k\ell$ respectively. Also, let $P_D$ denote the projection matrix of $D$, and hence if $D$ is a matrix with full column-rank then $P_D = D(D^T \cdot D)^{-1} \cdot D^T$. The $n \times n$ identity matrix is denoted as $I_n$, and for convenience of notation let $\tilde{X} = B \cdot X$. First we prove the following lemma which gives an expression for $D$ and $E$ if $\nabla_{\text{vec}(D^T)}\mathcal{L}(\overline{Y})$ and $\nabla_{\text{vec}(E^T)}\mathcal{L}(\overline{Y})$ are zero.

**Lemma C.1** (Derivatives with respect to $D$ and $E$).

1. $\nabla_{\text{vec}(D^T)}\mathcal{L}(\overline{Y}) = vec(r)^T(I_m \otimes (E \cdot \tilde{X})^T)$, and

2. $\nabla_{\text{vec}(E^T)}\mathcal{L}(\overline{X}) = vec(r)^T(D \otimes \tilde{X})^T$

*Proof.*    1. Since $\mathcal{L}(\overline{Y}) = \frac{1}{2}\text{vec}(r)^T \cdot \text{vec}(r)$,

$$
\begin{aligned}
\nabla_{\text{vec}(D^T)}\mathcal{L}(\overline{Y}) = \text{vec}(r)^T \cdot \nabla_{\text{vec}(D^T)}\text{vec}(r) &= \text{vec}(r)^T(\text{vec}_{(D^T)}(\tilde{X}^T \cdot E^T \cdot D^T)) \\
= \text{vec}(r)^T(I_m \otimes (E \cdot \tilde{X})^T) \cdot \nabla_{\text{vec}(D^T)}\text{vec}(D^T) &= \text{vec}(r)^T(I_m \otimes (E \cdot \tilde{X})^T)
\end{aligned}
$$

2. Similarly,

$$
\begin{aligned}
\nabla_{\text{vec}(E^T)}\mathcal{L}(\overline{Y}) = \text{vec}(r)^T \cdot \nabla_{\text{vec}(E^T)}\text{vec}(r) &= \text{vec}(r)^T(\text{vec}_{(E^T)}(\tilde{X}^T \cdot E^T \cdot D^T)) \\
= \text{vec}(r)^T(D \otimes \tilde{X}^T) \cdot \nabla_{\text{vec}(E^T)}\text{vec}(E^T) &= \text{vec}(r)^T(D \otimes \tilde{X}^T)
\end{aligned}
$$

$\square$

Assume the rank of $D$ is equal to $p$. Hence there is an invertible matrix $C \in \mathbb{R}^{k\times k}$ such that $\tilde{D} = D \cdot C$ is such that the last $k - p$ columns of $\tilde{D}$ are zero and the first $p$ columns of $\tilde{D}$ are linearly independent (via Gauss elimination). Let $\tilde{E} = C^{-1} \cdot E$. Without loss of generality it can be assumed $\tilde{D} \in \mathbb{R}^{d\times p}$, and $\tilde{E} \in \mathbb{R}^{p\times d}$, by restricting restricting $\tilde{D}$ to its first $p$ columns (as the remaining are

zero) and $\tilde{E}$ to its first $p$ rows. Hence, $\tilde{D}$ is a full column-rank matrix of rank $p$, and $DE = \tilde{D}\tilde{E}$. Claims C.1 and C.2 aid us in the completing the proof of the theorem. First the proof of theorem is completed using these claims, and at the end the two claims are proved.

**Claim C.1** (Representation at the critical point).

  1. $\tilde{E} = (\tilde{D}^T\tilde{D})^{-1}\tilde{D}^T Y \tilde{X}^T (\tilde{X} \cdot \tilde{X}^T)^{-1}$

  2. $\tilde{D}\tilde{E} = P_{\tilde{D}} Y \tilde{X}^T (\tilde{X} \cdot \tilde{X}^T)^{-1}$

**Claim C.2.**      1. $\tilde{E}B\tilde{D} = (\tilde{E}BY\tilde{X}^T\tilde{E}^T)(\tilde{E}\tilde{X}\tilde{X}^T\tilde{E}^T)^{-1}$

  2. $P_{\tilde{D}}\Sigma = \Sigma P_{\tilde{D}} = P_{\tilde{D}}\Sigma P_{\tilde{D}}$

We denote $\Sigma(B)$ as $\Sigma$ for convenience. Since $\Sigma$ is a real symmetric matrix, there is an orthogonal matrix $U$ consisting of the eigenvectors of $\Sigma$, such that $\Sigma = U \wedge U^T$, where $\wedge$ is a $m \times m$ diagonal matrix whose first $\ell$ diagonal entries are $\lambda_1, \ldots, \lambda_\ell$ and the remaining entries are zero. Let $u_1, \ldots, u_m$ be the columns of $U$. Then for $i \in [\ell]$, $u_i$ is the eigenvector of $\Sigma$ corresponding to the eigenvalue $\lambda_i$, and $\{u_{\ell+1}, \ldots, u_{d_y}\}$ are the eigenvectors of $\Sigma$ corresponding to the eigenvalue 0.

Note that $P_{U^T\tilde{D}} = U^T\tilde{D}(\tilde{D}^TU^TU\tilde{D})^{-1}\tilde{D}^TU = U^TP_{\tilde{D}}U$, and from part two of Claim C.2 we have

$$(UP_{U^T\tilde{D}}U^T)\Sigma = \Sigma(UP_{U^T\tilde{D}}U^T) \tag{9}$$

$$U \cdot P_{U^T\tilde{D}} \wedge U^T = U \wedge P_{U^T\tilde{D}}U^T \tag{10}$$

$$P_{U^T\tilde{D}}\wedge = \wedge P_{U^T\tilde{D}} \tag{11}$$

Since $P_{U^T\tilde{D}}$ commutes with $\wedge$, $P_{U^T\tilde{D}}$ is a block-diagonal matrix comprising of two blocks $P_1$ and $P_2$: the first block $P_1$ is an $\ell \times \ell$ diagonal block, and $P_2$ is a $(m - \ell) \times (m - \ell)$ matrix. Since $P_{U^T\tilde{D}}$ is orthogonal projection matrix of rank $p$ its eigenvalues are 1 with multiplicity $p$ and 0 with multiplicity $m - p$. Hence at most $p$ diagonal entries of $P_1$ are 1 and the remaining are 0. Finally observe that

$$\mathcal{L}(\overline{Y}) = \mathrm{tr}((\overline{Y} - Y)(\overline{Y} - Y)^T)$$
$$= \mathrm{tr}(YY^T) - 2\mathrm{tr}(\overline{Y}Y^T) + \mathrm{tr}(\overline{YY}^T)$$
$$= \mathrm{tr}(YY^T) - 2\mathrm{tr}(P_{\tilde{D}}\Sigma) + \mathrm{tr}(P_{\tilde{D}}\Sigma P_{\tilde{D}})$$
$$= \mathrm{tr}(YY^T) - \mathrm{tr}(P_{\tilde{D}}\Sigma)$$

The second line in the above equation follows using the fact that $\mathrm{tr}(\overline{Y}Y^T) = \mathrm{tr}(Y\overline{Y}^T)$, the third line in the above equation follows by substituting $\overline{Y} = P_{\tilde{D}}Y\tilde{X}^T \cdot (\tilde{X} \cdot \tilde{X}^T)^{-1} \cdot \tilde{X}$ (from part two of Claim C.1), and the last line follows from part two of Claim C.2. Substituting $\Sigma = U \wedge U^T$, and $P_{\tilde{D}} = UP_{U^T\tilde{D}}U^T$ in the above equation we have,

$$\mathcal{L}(\overline{Y}) = \mathrm{tr}(YY^T) - \mathrm{tr}(UP_{U^T\tilde{D}} \wedge U^T)$$
$$= \mathrm{tr}(YY^T) - \mathrm{tr}(P_{U^T\tilde{D}}\wedge)$$

The last line the above equation follows from the fact that $\mathrm{tr}(UP_{U^T\tilde{D}} \wedge U^T) = \mathrm{tr}(P_{U^T\tilde{D}} \wedge U^TU) = \mathrm{tr}(P_{U^T\tilde{D}}\wedge)$. From the structure of $P_{U^T\tilde{D}}$ and $\wedge$ it follows that there is a subset $I \subseteq [\ell]$, $|I| \le p$ such that $\mathrm{tr}(P_{U^T\tilde{D}}\wedge) = \sum_{i \in I} \lambda_i$. Hence, $\mathcal{L}(\overline{Y}) = \mathrm{tr}(YY^T) - \sum_{i \in I} \lambda_i$.

Since $P_{\tilde{D}} = UP_{U^T\tilde{D}}U^T$, there is a $p \times p$ invertible matrix $M$ such that

$$\tilde{D} = (U \cdot V)_{I'} \cdot M \;\; ,\text{and} \;\; \tilde{E} = M^{-1}(V^TU^T)_{I'}Y\tilde{X}^T(\tilde{X}\tilde{X}^T)^{-1}$$

where $V$ is a block-diagonal matrix consisting of two blocks $V_1$ and $V_2$: $V_1$ is equal to $I_\ell$, and $V_2$ is an $(m - \ell) \times (m - \ell)$ orthogonal matrix, and $I'$ is such that $I \subseteq I'$ and $|I'| = p$. The relation for $\tilde{E}$ in the above equation follows from part one of Claim C.1. Note that if $I' \subseteq [\ell]$, then $I = I'$, that is $I$ consists of indices corresponding to eigenvectors of non-zero eigenvalues.

Recall that $\tilde{D}$ was obtained by truncating the last $k - p$ zero rows of $DC$, where $C$ was a

$k \times k$ invertible matrix simulating the Gaussian elimination. Let $[M|O_{p\times(k-p)}]$ denoted the $p \times k$ matrix obtained by augmenting the columns of $M$ with $(k-p)$ zero columns. Then

$$D = (UV)_{I'}[M|O_{p\times(k-p)}]C^{-1} \ .$$

Similarly, there is a $p \times (k-p)$ matrix $N$ such that

$$E = C[\tfrac{M^{-1}}{N}]((UV)_{I'})^T Y \tilde{X}^T (\tilde{X}\tilde{X}^T)^{-1}$$

where $[\tfrac{M^{-1}}{N}]$ denotes the $k \times p$ matrix obtained by augmenting the rows of $M^{-1}$ with the rows of $N$. Now suppose $I \neq [k]$, and hence $I' \neq [k]$. Then we will show that there are matrices $D'$ and $E'$ arbitrarily close to $D$ and $E$ respectively such that if $Y' = D'E'\tilde{X}$ then $\mathcal{L}(Y') < \mathcal{L}(\overline{Y})$. There is an $a \in [k] \setminus I'$, and $b \in I'$ such that $\lambda_a > \lambda_b$ ($\lambda_b$ could also be zero). Denote the columns of the matrix $UV$ as $\{v_1, \ldots, v_m\}$, and observe that $v_i = u_i$ for $i \in [\ell]$ (from the structure of $V$). For $\epsilon > 0$ let $u'_b = (1+\epsilon^2)^{-\frac{1}{2}}(v_b + \epsilon u_a)$. Define $U'$ as the matrix which is equal to $UV$ except that the column vector $v_b$ in $UV$ is replaced by $u'_b$ in $U'$. Since $a \in [k] \subseteq [\ell]$ and $a \notin I'$, $v_a = u_a$ and $(U'_{I'})^T U'_{I'} = I_p$. Define

$$D' = U'_{I'}[M|O_{p\times(k-p)}]C^{-1} \ , \text{ and } \ E' = C[\tfrac{M^{-1}}{N}](U'_{I'})^T Y \tilde{X}^T (\tilde{X}\tilde{X}^T)^{-1}$$

and let $Y' = D'E'\tilde{X}$. Now observe that, $D'E' = U'_{I'}(U_{I'})^T Y \tilde{X}^T (\tilde{X}\tilde{X}^T)^{-1}$, and that

$$\mathcal{L}(Y') = \text{tr}(YY^T) - \sum_{i \in I} \lambda_i - \frac{\epsilon^2}{1+\epsilon^2}(\lambda_a - \lambda_b) = \mathcal{L}(\overline{Y}) - \frac{\epsilon^2}{1+\epsilon^2}(\lambda_a - \lambda_b)$$

Since $\epsilon$ can be set arbitrarily close to zero, it can be concluded that there are points in the neighbourhood of $\overline{Y}$ such that the loss at these points are less than $\mathcal{L}(\overline{Y})$. Further, since $\mathcal{L}$ is convex with respect to the parameters in $D$ (respectively $E$), when the matrix $E$ is fixed (respectively $D$ is fixed) $\overline{Y}$ is not a local maximum. Hence, if $I \neq [k]$ then $\overline{Y}$ represents a saddle point, and in particular $\overline{Y}$ is local/global minima if and only if $I = [k]$.

*Proof of Claim C.1.* Since $\nabla_{\text{vec}(E^T)}\mathcal{L}(\overline{X})$ is equal to zero, from the second part of Lemma C.1 the following holds,

$$\tilde{X}(Y - \overline{Y})^T D = \tilde{X}Y^T D - \tilde{X}\overline{Y}^T D = 0$$
$$\Rightarrow \tilde{X}\tilde{X}^T E^T D^T D = \tilde{X}Y^T D$$

Taking transpose on both sides

$$\Rightarrow D^T DE\tilde{X}\tilde{X}^T = D^T Y \tilde{X}^T \tag{12}$$

Substituting $DE$ as $\tilde{D}\tilde{E}$ in Equation 12, and multiplying Equation 12 by $C^T$ on both the sides from the left, Equation 13 follows.

$$\Rightarrow \tilde{D}^T \tilde{D}\tilde{E}\tilde{X}\tilde{X}^T = \tilde{D}^T Y \tilde{X}^T \tag{13}$$

Since $\tilde{D}$ is full-rank, we have

$$\tilde{E} = (\tilde{D}^T \tilde{D})^{-1}\tilde{D}^T Y \tilde{X}^T (\tilde{X}\tilde{X}^T)^{-1}. \tag{14}$$

and,

$$\tilde{D}\tilde{E} = P_{\tilde{D}}Y \tilde{X}^T (\tilde{X}\tilde{X}^T)^{-1} \tag{15}$$

$\square$

*Proof of Claim C.2.* Since $\nabla_{\text{vec}(D^T)}\mathcal{L}(\overline{Y})$ is zero, from the first part of Lemma C.1 the following holds,

$$E\tilde{X}(Y - \overline{Y})^T = E\tilde{X}Y^T - E\tilde{X} \cdot \overline{Y}^T = 0$$
$$\Rightarrow E\tilde{X}\tilde{X}^T E^T D^T = E\tilde{X}Y^T \tag{16}$$

Substituting $E^T \cdot D^T$ as $\tilde{E}^T \cdot \tilde{D}^T$ in Equation 12, and multiplying Equation 16 by $C^{-1}$ on both the sides from the left Equation 17 follows.

$$\tilde{E}\tilde{X}\tilde{X}^T \tilde{E}^T \tilde{D}^T = \tilde{E}\tilde{X}Y^T \tag{17}$$

Taking transpose of the above equation we have,

$$\tilde{D}\tilde{E}\tilde{X}\tilde{X}^T\tilde{E}^T = Y\tilde{X}^T\tilde{E}^T \tag{18}$$

From part 1 of Claim C.1, it follows that $\tilde{E}$ has full row-rank, and hence $\tilde{E}\tilde{X}\tilde{X}^T\tilde{E}^T$ is invertible. Multiplying the inverse of $\tilde{E}\tilde{X}\tilde{X}^T\tilde{E}^T$ from the right on both sides and multiplying $\tilde{E}B$ from the left on both sides of the above equation we have,

$$\tilde{E}B\tilde{D} = (\tilde{E}BY\tilde{X}^T\tilde{E}^T)(\tilde{E}\tilde{X}\tilde{X}^T\tilde{E}^T)^{-1} \tag{19}$$

This proves part one of the claim. Moreover, multiplying Equation 18 by $\tilde{D}^T$ from the right on both sides

$$\tilde{D}\tilde{E}\tilde{X}\tilde{X}^T\tilde{E}^T\tilde{D}^T = Y\tilde{X}^T\tilde{E}^T\tilde{D}^T$$
$$\Rightarrow (P_{\tilde{D}}Y\tilde{X}^T(\tilde{X}\tilde{X}^T)^{-1})(\tilde{X}\tilde{X}^T)((\tilde{X}\tilde{X}^T)^{-1}\tilde{X}Y^T P_{\tilde{D}}) = Y\tilde{X}^T((\tilde{X}\tilde{X}^T)^{-1}\tilde{X}Y^T \cdot P_{\tilde{D}})$$
$$\Rightarrow P_{\tilde{D}}Y\tilde{X}^T(\tilde{X}\tilde{X}^T)^{-1}\tilde{X}Y^T P_{\tilde{D}} = Y\tilde{X}^T(\tilde{X}\tilde{X}^T)^{-1}\tilde{X}Y^T \cdot P_{\tilde{D}}$$

The second line the above equation follows by substituting $\tilde{D}\tilde{E} = P_{\tilde{D}}Y\tilde{X}^T(\tilde{X}\tilde{X}^T)^{-1}$ (from part 2 of Claim C.1). Substituting $\Sigma = Y\tilde{X}^T(\tilde{X}\tilde{X}^T)^{-1}\tilde{X}Y^T$ in the above equation we have

$$P_{\tilde{D}}\Sigma P_{\tilde{D}} = \Sigma \cdot P_{\tilde{D}}$$

Since $P_{\tilde{D}}^T = P_{\tilde{D}}$, and $\Sigma^T = \Sigma$, we also have $\Sigma P_{\tilde{D}} = P_{\tilde{D}}\Sigma$. □

# D ADDITIONAL TABLES AND PLOTS FROM SECTION 6

## D.1 PLOTS FROM SECTION 6.1

Figure 7 displays the number of parameters in the dense linear layer of the original model and in the replaced butterfly based network. Figure 9 reports the results for the NLP tasks done as part of experiment in Section 6.1. Figure 8 displays the number of parameter in the original model and the butterfly model. Figures 10 and 11 reports the training and inference times required for the original model and the butterfly model in each of the experiments. The training and and inference times in Figures 10 and 11 are averaged over 100 runs. Figure 12 is the same as the right part of Figure 1 but here we compare the test accuracy of the original and butterfly model for the the first 20 epochs.

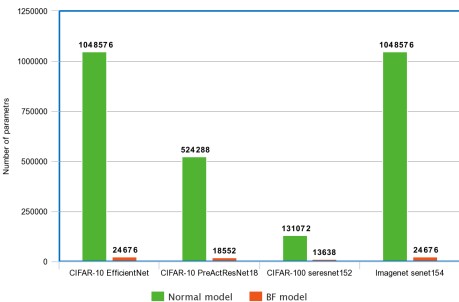 

Figure 7: Number of parameters in the dense linear layer of the original model and in the replaced butterfly based architecture; Left: Vision data, Right: NLP

## D.2 PLOTS FROM SECTION 6.2

**Data Matrices**: The data matrices are as in Table 2. Gaussian 1 and Gaussian 2 are Gaussian matrices with rank 32 and 64 respectively. Rank $r$ Gaussian matrices are constructed as follows: $r$ orthonormal vectors of size 1024 are sampled at random and the columns of the matrix are random linear combinations of these vectors determined by choosing the coefficients independently and uniformly at random from the Gaussian distribution with mean 0 and variance 0.01. The data matrix for MNIST is constructed as follows: each row corresponds to an image represented as a 28 ×

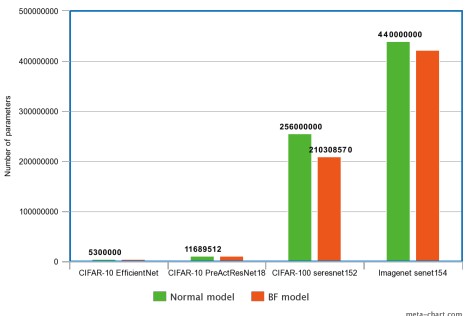 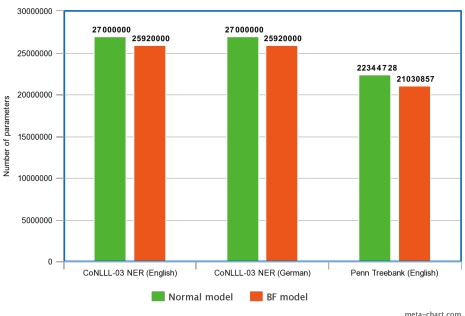

Figure 8: Total number of parameters in the original model and the butterfly model; Left: Vision data, Right: NLP

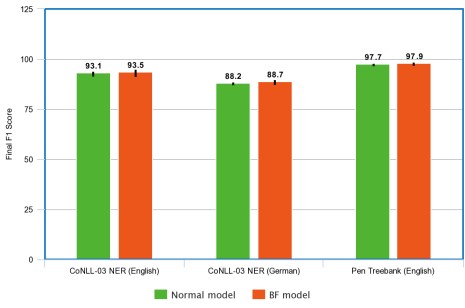 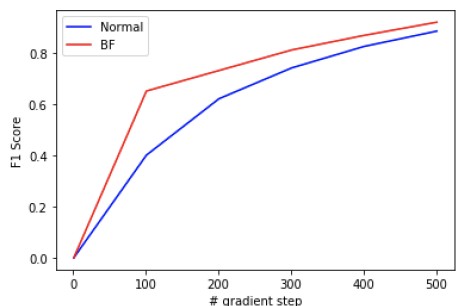

Figure 9: Right: Final F1 Score for different NLP models and data sets. Left: F1 comparison in the first few epochs with different models on CoNLL-03 Named Entity Recognition (English) with the flair's Sequence Tagger

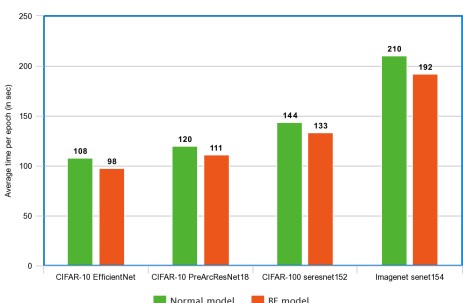 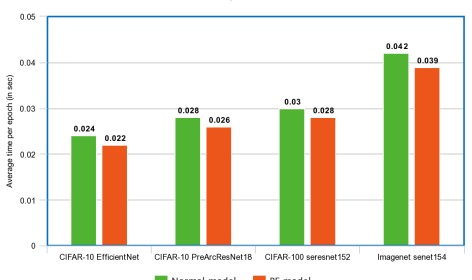

Figure 10: Training/Inference times for Vision Data; Left: Training time, Right: Inference time

28 matrix (pixels) sampled uniformly at random from the MNIST database of handwritten digits (LeCun and Cortes, 2010) which is extended to a $32 \times 32$ matrix by padding numbers close to zero and then represented as a vector of size $1024$ in column-first ordering[8]. Similar to the MNIST every row of the data matrix for Olivetti corresponds to an image represented as a $64 \times 64$ matrix sampled uniformly at random from the Olivetti faces data set (Cambridge, 1994), which is represented as a vector of size $4096$ in column-first ordering. Finally, for HS-SOD the data matrix is a $1024 \times 768$ matrix sampled uniformly at random from HS-SOD – a dataset for hyperspectral images from natural scenes (Imamoglu et al., 2018).

Figure 13 reports the losses for the Gaussian 2, Olivetti, and Hyper data matrices.

---

[8]Close to zero entries are sampled uniformly at random according to a Gaussian distribution with mean zero and variance $0.01$.

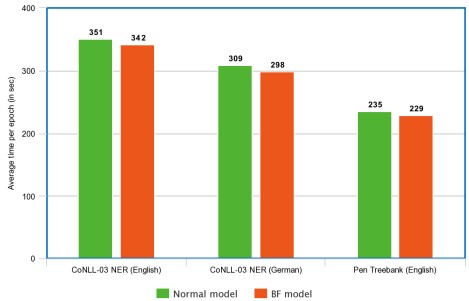
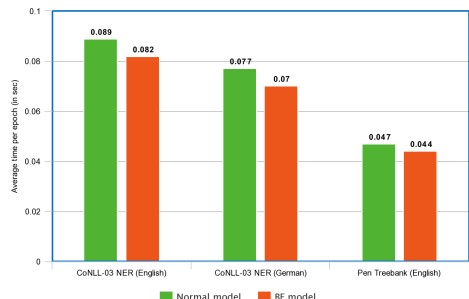

Figure 11: Training/Inference times for NLP; Left: Training time, Right: Inference time

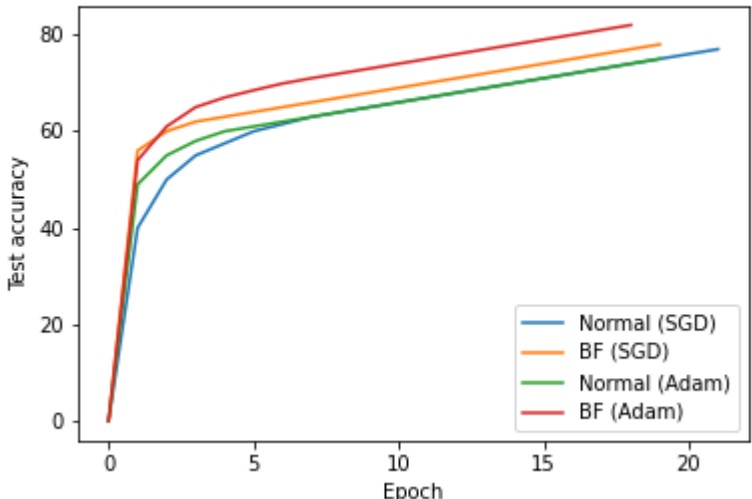

Figure 12: Comparison of test accuracy in the first 20 epochs with different models and optimizers on CIFAR-10 with PreActResNet18

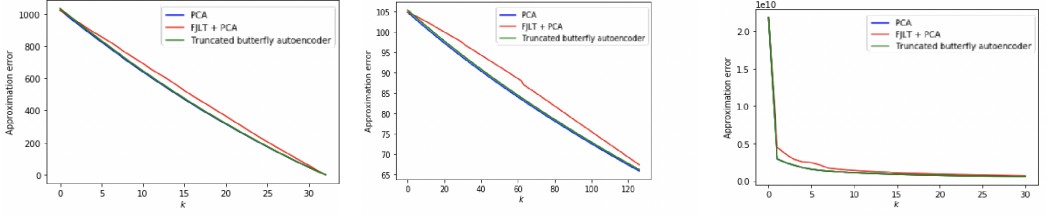

Figure 13: Approximation error on data matrix with various methods for various values of $k$. From left to right: Gaussian 2, Olivetti, Hyper

## E  MISSING PLOTS FROM SECTION 7

In this section we state a few additional cases that were done as part of the experiment in Section 7. Figure 14 compares the test errors of the different methods in the extreme case when $k = 1$. Figure 15 compares the test errors of the different methods for various values of $\ell$. Figure 16 shows the test error for $\ell = 20$ and $k = 10$ during the training phase on HS-SOD. Observe that the butterfly

learned is able to surpass sparse learned after a merely few iterations. Finally Table 4 compares the test error for different values of $\ell$ and $k$.

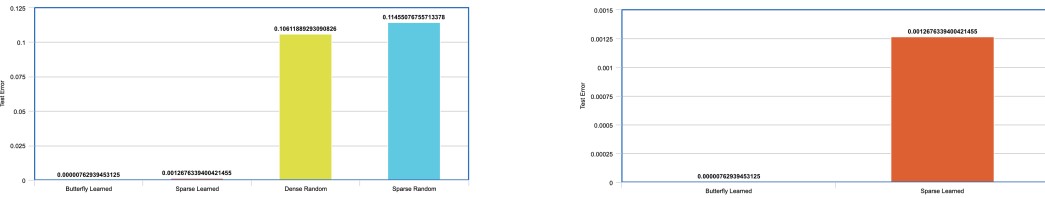

Figure 14: Test errors on HS-SOD for $\ell = 20$ and $k = 1$, zoomed on butterfly and sparse learned in the right

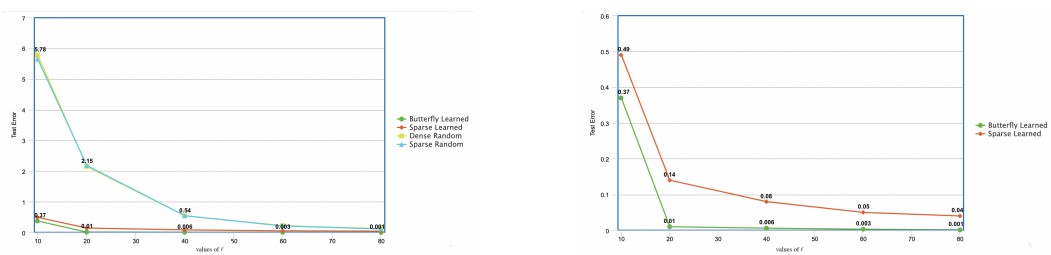

Figure 15: Test error when $k = 10$, $\ell = [10, 20, 40, 60, 80]$ on HS-SOD, zoomed on butterfly and sparse learned in the right

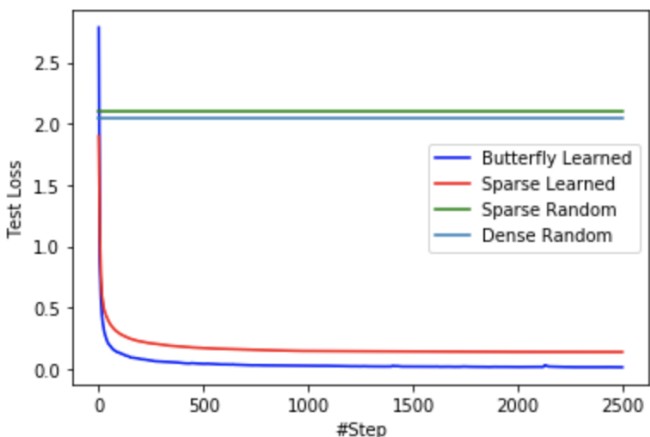

Figure 16: Test error when $k = 10$, $\ell = 20$ during the training phase on HS-SOD

## F  BOUND ON NUMBER OF EFFECTIVE WEIGHTS IN TRUNCATED BUTTERFLY NETWORK

A butterfly network for dimension $n$, which we assume for simplicity to be an integral power of 2, is $\log n$ layers deep. Let $p$ denote the integer $\log n$. The set of nodes in the first (input) layer will be denoted here by $V^{(0)}$. They are connected to the set of $n$ nodes $V^{(1)}$ from the next layer, and so on until the nodes $V^{(p)}$ of the output layer. Between two consecutive layers $V^{(i)}$ and $V^{(i+1)}$, there are $2n$ weights, and each node in $V^{(i)}$ is adjacent to exactly two nodes from $V^{(i+1)}$.

When truncating the network, we discard all but some set $S^{(p)} \subseteq V^{(p)}$ of at most $\ell$ nodes in the last layer. These nodes are connected to a subset $S^{(p-1)} \subseteq V^{(p-1)}$ of at most $2\ell$ nodes from the

| k, $\ell$, Sketch | Hyper | Cifar-10 | Tech |
|---|---|---|---|
| 1, 5, Butterfly | **0.0008** | **0.173** | **0.188** |
| 1, 5, Sparse | 0.003 | 1.121 | 1.75 |
| 1, 5, Random | 0.661 | 4.870 | 3.127 |
| 1, 10, Butterfly | **0.0002** | **0.072** | **0.051** |
| 1, 10, Sparse | 0.002 | 0.671 | 0.455 |
| 1, 10, Random | 0.131 | 1.82 | 1.44 |
| 10, 10, Butterfly | **0.031** | **0.751** | **0.619** |
| 10, 10, Sparse | 0.489 | 6.989 | 7.154 |
| 10, 10, Random | 5.712 | 26.133 | 18.805 |
| 10, 20, Butterfly | **0.012** | **0.470** | **0.568** |
| 10, 20, Sparse | 0.139 | 3.122 | 3.134 |
| 10, 20, Random | 2.097 | 9.216 | 8.22 |
| 10, 40, Butterfly | **0.006** | | **0.111** |
| 10, 40, Sparse | 0.081 | | 0.991 |
| 10, 40, Random | 0.544 | | 3.304 |
| 20, 20, Butterfly | **0.058** | | **1.38** |
| 20, 20, Sparse | 0.229 | | 8.14 |
| 20, 20, Random | 4.173 | | 15.268 |
| 20, 40, Butterfly | **0.024** | | **0.703** |
| 20, 40, Sparse | 0.247 | | 3.441 |
| 20, 40, Random | 1.334 | | 6.848 |
| 30, 30, Butterfly | **0.027** | | **1.25** |
| 30, 30, Sparse | 0.749 | | 7.519 |
| 30, 30, Random | 3.486 | | 13.168 |
| 30, 60, Butterfly | **0.014** | | **0.409** |
| 30, 60, Sparse | 0.331 | | 2.993 |
| 30, 60, Random | 2.105 | | 5.124 |

Table 4: Test error for different $\ell$ and $k$

preceding layer using at most $2\ell$ weights. By induction, for all $i \geq 0$, the set of nodes $S^{(p-i)} \subseteq V^{(p-i)}$ is of size at most $2^i \cdot \ell$, and is connected to the set $S^{(p-i-1)} \subseteq V^{(p-i-1)}$ using at most $2^{i+1} \cdot \ell$ weights.

Now take $k = \lceil \log_2(n/\ell) \rceil$. By the above, the total number of weights that can participate in a path connecting some node in $S^{(p)}$ with some node in $V^{(p-k)}$ is at most

$$2\ell + 4\ell + \cdots + 2^k \ell \leq 4n .$$

From the other direction, the total number of weights that can participate in a path connecting any node from $V^{(0)}$ with any node from $V^{(p-k)}$ is $2n$ times the number of layers in between, or more precisely:

$$2n(p - k) = 2n(\log_2 n - \lceil \log_2(n/\ell) \rceil) \leq 2n(\log_2 n - \log_2(n/\ell) + 1) = 2n(\log \ell + 1) .$$

The total is $2n \log \ell + 6n$, as required.

