# OpenReview forum: "Sparse Linear Networks with a Fixed Butterfly Structure: Theory and Practice"
_ICLR.cc/2021/Conference — Reject_

### Official Review · AnonReviewer2 · 2020-10-20
**Further details on experimental setting are required**

**Rating:** 6
**Confidence:** 3

**Review:**

This paper proposes to impose a particular sparsity structure (butterfly network) to replace dense connected layers in deep neural networks. It is motivated by the theoretical results involving the Fast Johnson Lindenstrauss Transform (FJLT). The work is well motivated and experimental results validate the theoretical findings. However, it is not clear the advantage for the case of image datasets as CIFAR10 and CIFAR100. I have the following comments and questions that should be clarified to evaluate the relevance of the results:
-	When comparing with other architectures on image classification tasks (CIFAR10 and CIFAR 100), what is the dense layer that is replaced by the butterfly structure? Most very well-known architectures are based on large concatenation of convolutional layers with a dense layer at the end. Is this last layer the one that is replaced? What is the compression attained by this replacement?
-	In this architecture, the sparsity pattern is fixed, and it seems to work very well for the dataset used in the paper. It would be interesting to provide some insights on why this particularly structure works well on natural data. Is there any property on datasets that makes butterfly pattern optimal?
-	For natural images, it is well known that sparsity structure imposed by convolutional layers is optimal because of local structure of natural images. I think a comparison replacing a convolutional layer by a butterfly layer could bring some useful insights.
-	Theorem 1: What is Omega in the exponent? I couldn’t find its definition.
-	Figure 1(a): The small difference between normal and BF models seems not to be statistically significant. It would be good to show some variability of results (error bars).
-	 For the results of Figure 1(b), it would be also useful to report the training time reduction and model compression attained by the normal and BF models.
-	Figure 1(b): The starting point for the four methods should be the same. Please include in the plot the Test Accuracy at the beginning (epoch 0).
-	Figure 1(b): Please extend the plots beyond epoch 20 to see how the values are stabilized for the four methods.
-	Quality of Figures, for example Fig 3, must be improved (too small fonts, blurred, etc.)

---

> ### Author Response · Authors · 2020-11-17
> **Responses to the Queries Raised by the Reviewer**
>
> We appreciate the significant effort of all the reviewers to deliver such detailed and constructive suggestions. We also appreciate that most reviewers identify the novelty of our work. We have prepared a new version of the document by incorporating all the suggestions. Specific queries raised by Reviewer 4 are addressed below.
>
> 1.  When comparing with other architectures on image classification tasks (CIFAR10 and CIFAR 100), what is the dense layer that is replaced by the butterfly structure? Most very well-known architectures are based on large concatenation of convolutional layers with a dense layer at the end. Is this last layer the one that is replaced? What is the compression attained by this replacement?
>
> Response: Yes, in the experiments in Section 6.1 the last layer in all the considered architectures is replaced. We state this in footnote 5, page 6 (Also see response to point 3 of reviewer 3). We have added the exact parameter counts in Figures 7 and 8 in Appendix D.1 of our new version.
>
> 2.  In this architecture, the sparsity pattern is fixed, and it seems to work very well for the dataset used in the paper. It would be interesting to provide some insights on why this particular structure works well on natural data. Is there any property on datasets that makes butterfly pattern optimal?
>
> Response: We note that the butterfly structure may work well not only on natural data, but also on synthetic data. This is supported by the extensive known theory of approximating matrices with the exact same structure.
>
> 3. For natural images, it is well known that sparsity structure imposed by convolutional layers is optimal because of local structure of natural images. I think a comparison replacing a convolutional layer by a butterfly layer could bring some useful insights.
>
> Response: We include it as a possible future direction. We note that a convolutional layer is suitable in data which is represented in a basis that is induced by temporal or spatial structure, while our architecture works with any basis.
>
> 4. Theorem 1: What is Omega in the exponent? I couldn’t find its definition.
>
> Response: We have renamed Theorem 1 as Proposition 1. In this \Omega is the standard ‘big Omega’ notation. A note about this notation can be found here https://www.freecodecamp.org/news/big-omega-notation/
>
> 5. Figure 1(a): The small difference between normal and BF models seems not to be statistically significant. It would be good to show some variability of results (error bars). For the results of Figure 1(b), it would be also useful to report the training time reduction and model compression attained by the normal and BF models.
>
> Response: We have made the changes to Figure 1a, and added the training times and the number of parameters in the new version of our document. See Figures 7, 8, 10, 11 in Appendix D.1
>
> 6.  Figure 1(b): The starting point for the four methods should be the same. Please include in the plot the Test Accuracy at the beginning (epoch 0). Figure 1(b): Please extend the plots beyond epoch 20 to see how the values are stabilized for the four methods. Quality of Figures, for example Fig 3, must be improved (too small fonts, blurred, etc.)
>
> Response: We have incorporated the changes in the new version of our document. The plot for 20 epochs has been added in Figure 12, Appendix D.1.

---

> > ### Comment · AnonReviewer2 · 2020-11-17
> > **feedback on responses**
> >
> > Dear Authors,
> > Thank you very much for providing answer to my questions. Most of my questions were satisfactory answered, however some were not totally clarified, for example:
> > - Your answer to question 2: "We note that the butterfly structure may work well not only on natural data, but also on synthetic data. This is supported by the extensive known theory of approximating matrices with the exact same structure". No response about natural data properties that make the butterfly structure to work well. Moreover, it is mentioned that there is extensive known theory of approximating matrices with the same structure but there is no reference to such papers or literature.
> > -  Response to question 5: Authors made changes on Fig. 1a. It seems error bars were added but I couldn’t find what kind of errors are shown. Are they standard errors? Or standard deviations? Without that specification it is not possible to interpret the results.
> > I found the work interesting and I hope the authors could improve a little bit the paper in its final version. I decided to raise the score.

---

> > > ### Author Response · Authors · 2020-11-23
> > > **Responses to the Queries**
> > >
> > > We thank the reviewer for the additional feedback.
> > >
> > > With respect to Fig1a, we have added a line in the new version stating that the black vertical lines denote the error bars corresponding to standard deviation, and the values above the rectangles denote the average accuracy.
> > >
> > > With respect to the first point, we are sorry for not responding clearly to your question.  We feel that we may have misunderstood your concern, and appreciate your time reading our rather detailed response herein.
> > >
> > > On the properties of natural data that make the butterfly structure work well: We do not know what it is about 'natural data' that makes the butterfly structure work well.  From the experimental side, we see that our approach seems promising on 'natural data'.  From the theoretical side, we know that even if we randomly permute the coordinates of the input data, thus breaking any natural temporal/spatial structure, the results would hold, because the temporal/spatial structure is not used in the analysis in any way.  In that sense our architecture is different from (say) convolutional layers, which are also sparse, and strongly rely on the temporal/spatial structure.  On the other hand, we do not claim that our method can be used as a replacement to CNN's, because obviously we do want to take advantage of the temporal/spatial structure of the signals when available.  In fact, our method is much more suitable for replacing dense layers in places where, incidentally, one would not think of putting a CNN.  Also incidentally, these dense layers can be bottleneck (in terms of computational resources) in training and testing.
> > >
> > > Regarding literature on the extensive known theory of approximating matrices with the same structure:  The references in the literature are indeed mentioned in the following papers, which are readily cited in our work. Ailon+Chazelle 2009, Ailon+Liberty 2009, Clarkson+Woodruff 2009, Krahmer+Ward 2011, Sarlos 2006. Some classic theory on which these works are based on is Johnson+Lindenstrauss 1984 (this is before the discovery of the fast Johnson-Lindenstrauss trasform, which gives rise to the butterfly structure).
> > >
> > > We also cite some related theory on fast, low rank matrix approximation: Indyk+Vakilian+Yuan 2019, Pillai+Smith 2020 (although they do not use the butterfly structure, they use something else, and our experiments use their data).
> > >
> > > Regarding our remark in the first rebuttal: "We note that the butterfly structure may work well not only on natural data, but also on synthetic data" - By this we meant, as stated above here, that there is nothing in the theory that uses any property of "nature", and therefore we expect this scheme to work on any data, including synthetic.  In fact we even expect the method to "work" on contrived, deliberately crafted data in the sense that our approach would be, in terms of accuracy, on par with an architecture that uses dense layers naively, except that it would probably require fewer resources.  As a note: We would probably have to assume that the contrived data does not "know" about the initialization of the network, because with that knowledge it is possible to generate data that falls in the kernel of the dimension reducing linear transformation (the "truncated butterfly") , causing complete collapse of information.  But this is exponentially unlikely if the contrived data generation is unaware of the initialization.

---

### Official Review · AnonReviewer4 · 2020-10-25
**Motivated by Johnson-Lindenstrauss-type results, this paper proposes to replace a dense linear layer in any neural network by a composition of two truncated butterfly networks and a smaller dense layer in between. Theoretical result is given for the two-layer encoder-decoder network where the encoder is replaced by one truncated butterfly network and a dense layer. Empirical results are given for several synthetic  and real datasets.**

**Rating:** 5
**Confidence:** 3

**Review:**

I list the strong and weak points in the following:

### Strong points

- It is a novel idea (only to my knowledge) to combine the butterfly network and FJLT to optimize the neural network architectures. Specifically, the proposed method replaces a dense linear layer by a composition of three layers, which are smaller and can be computed faster by FJLT.


### Weak points

- The paper says that the proposed method would speed up the training. But there is no quantitative argument. Also the paper does not report the computing time or memory used in all the experiments.

- In Definition 4.1, $n$ is required to be a power of 2. And in the experiments, it seems that all the $n$ are powers of 2 (except Tech). What do one do if $n$ is not a power of 2?

- In all the experiments, the final hidden layer is replaced by the proposed architecture. Is there any reason for doing this? How should one choose which layer(s) to be replaced?

- It is argued that by replacing a dense layer with a truncated butterfly network and a dense layer, the number of parameters is reduced from $kn$ to $k\ell+O(n\log\ell)$. But since the Big O can hide potentially large constants, I wonder how many parameters are used exactly. I expect the paper would report the parameter numbers in the experiments.

- Theorem 1 shows that replacing a dense layer by two truncated butterfly networks and a dense layer will not be too different from the original network. But this holds only when $J_1$ and $J_2$ are sampled from the FJLT distribution. Since the weights $J_1$ and $J_2$ are updated through the training process, the theorem will only hold at initialization. Then what is the point of Theorem 1?

---

> ### Author Response · Authors · 2020-11-17
> **Responses to the Queries Raised by the Reviewer**
>
> We appreciate the significant effort of all the reviewers to deliver such detailed and constructive suggestions. We also appreciate that most reviewers identify the novelty of our work. We have prepared a new version of the document by incorporating all the suggestions. Specific queries raised by Reviewer 3 are addressed below.
>
> 1. The paper says that the proposed method would speed up the training. But there is no quantitative argument. Also the paper does not report the computing time or memory used in all the experiments.
>
> Response: We have added the training times and the parameter count to the new version.  See Figures 7, 8, 10, 11 in Appendix D.1.
>
> 2. In Definition 4.1,  n is required to be a power of 2. And in the experiments, it seems that all the n are powers of 2 (except Tech). What do one do if  nis not a power of 2?
>
> Response: Suppose n is not a power of 2 and let n’ be the closest number to n that is greater than n and is a power of 2. Note that n’<2n. Now take the truncated butterfly network of \ell \times n’ and work with only the first n columns of this truncated butterfly network. We have added this as a footnote (page 4) in the new version of the document.
>
> 3. In all the experiments, the final hidden layer is replaced by the proposed architecture. Is there any reason for doing this? How should one choose which layer(s) to be replaced?
>
> Response: We experimented with the final layer. But we believe that multiple layers can be replaced simultaneously, and this is part of our ongoing research.
>
> 4. It is argued that by replacing a dense layer with a truncated butterfly network and a dense layer, the number of parameters is reduced from kn to kl+O(nlog⁡ l). But since the Big O can hide potentially large constants, I wonder how many parameters are used exactly. I expect the paper would report the parameter numbers in the experiments.
>
> Response: The effective number of parameters in a \ell x n truncated butterfly network is at most (2n\log \ell +6n). In the new version of our paper, we state this in the paragraph after Definition 4.1 (page 4), and include a proof of this fact in Appendix G for completeness.  We also state the exact parameters counts in Figures 7 and 8 in Appendix D.1.
>
> 5. Theorem 1 shows that replacing a dense layer by two truncated butterfly networks and a dense layer will not be too different from the original network. But this holds only when J_1 and J_2 are sampled from the FJLT distribution. Since the weights J_1 and J_2 are updated throughout the training process, the theorem will only hold at initialization. Then what is the point of Theorem 1?
>
> Response: Theorem 1, which is now proposition 1 (in the new version), shows that the new network has very little loss in representation compared to the old network with more parameters. In particular, it tells us that any solution in the original dense network is approximated w.h.p. in the modified sparse network, but as the reviewer adequately points out, it doesn't say what happens along the optimization path. Our experiments do indicate that learning is better and faster in the modified sparse network, and we leave the question of theoretically explaining that for future work.

---

### Official Review · AnonReviewer1 · 2020-10-29
**Interesting paper**

**Rating:** 7
**Confidence:** 5

**Review:**


The paper provides an interesting and novel use of butterfly factoziations in encoder-decoder networks. Specifically, the paper proposes replacing the
encoder with a truncated butterfly network followed by a dense linear layer. The parameters are chosen so as to keep the number of weights in the (replaced) encoder near linear in the input dimension. The authors provide a theoretical result related to auto-encoder optimization.

######

I vote for accepting the paper. The main reason is for my vote is that the main idea introduced is novel and the algorithmic contribution is substantial.

######

pros
+ The proposed truncated butterfly network is novel. Aside from the algorithmic contribution, the theorem in the paper raise important questions about
the optimization landscape of butterfly networks.
+ The paper is clearly written and well justified
+ Exhaustive literature survey and background on relevant work in both the matrix factorization and neural networks front

######

cons
- The beginning of section 7 needs to be expanded with more details on the original Indyk et al 2019 paper -- the authors mention that they use the same setting as
the 2019 paper but the details on how Indyk et al train their network are lacking without looking up the original paper.

- I think that low matrix approximation experimentation (Sec 7) can be more thorough. Why are only three datasets (in Table 3) used? Additionally and more importantly, Table 4 shows the approximation results for low values of k (max of 30) -- what happens when k=(min(n,d))? Also, the error is measured with respect to the best rank k approximation of X (the eq. following eq. 20) While this way of measuring the error is often used in the low rank matrix approximation literature it is not very informative of the actual approximation quality of \tilde X. It could well be the case that \tilde X = X_k but the error ||X-\tilde X||_F/||X||_F might  be poor hence this way of measuring the approximation performance is a better indicator of how well X is actually approximated in practice.

######

questions: see the above section

---

> ### Author Response · Authors · 2020-11-17
> **Responses to the Queries Raised by the Reviewer**
>
> We appreciate the significant effort of all the reviewers to deliver such detailed and constructive suggestions. We also appreciate that most reviewers identify the novelty of our work. We have prepared a new version of the document by incorporating all the suggestions. Specific queries raised by Reviewer 2 are addressed below.
>
> 1. The beginning of section 7 needs to be expanded with more details on the original Indyk et al 2019 paper -- the authors mention that they use the same setting as the 2019 paper but the details on how Indyk et al train their network are lacking without looking up the original paper.
>
> Response: Due to the limit of eight pages we tried to convey only the main idea and presented the results. We have used the additional page to add the necessary details to improve the readability of this section. We have included a line on how  [IndyVY19] train their network.
>
> 2. I think that low matrix approximation experimentation (Sec 7) can be more thorough.
>
> a) Why are only three datasets (in Table 3) used?
>
> Response a) The two datasets (HS-SOD and Tech) are used in [IndykVY19]. They have three more data sets generated from three videos on youtube respectively. Also, we feel the message and the utility of a truncated butterfly network is conveyed with the results established on the three datasets considered by us.
>
> b)Additionally and more importantly, Table 4 shows the approximation results for low values of k (max of 30) -- what happens when k=(min(n,d))?
>
> Response:b) Firstly, in low rank matrix recovery the interesting regime is k<< n, and we show results for this interesting regime. That is the approximation error is only going to improve for large k. In particular for k=(min(n,d)) the error would be zero. Secondly, the results in [IndykVY19] are also established for k at most 30.
>
> c) Also, the error is measured with respect to the best rank k approximation of X (the eq. following eq. 20) While this way of measuring the error is often used in the low rank matrix approximation literature it is not very informative of the actual approximation quality of \tilde X. It could well be the case that \tilde X = X_k but the error ||X-\tilde X||_F/||X||_F might be poor hence this way of measuring the approximation performance is a better indicator of how well X is actually approximated in practice.
>
> Response c): As remarked this is often used in low-rank matrix approximation, and also used in [IndyVY19]. But as noted, the actual error  ||X-\overline{X}||_F might be poor. This happens if ||X-X_k|| is large to begin with. But the || X-X_k|| is theoretically the best possible error that can be achieved (PCA), and in particular X_k is theoretically the best possible low-rank matrix recovery. Hence APP_{Te} is the best average expected error that one can hope for. This motivates why the error is measured with respect to the best rank k approximation of X.

---

### Official Review · AnonReviewer3 · 2020-10-31
**The paper studies the idea of butterfly networks, drawing inspiration from the sketching literature.**

**Rating:** 5
**Confidence:** 5

**Review:**

The paper studies “butterfly networks”, where, a logarithmic number of linear layers with sparse connections resembling the butterfly structure of the FFT algorithm, along with linear layers in smaller dimensions are used to approximate linear layers in larger dimensions. In general, the paper follows the idea of sketching to design new architectures that can reduce the number of trainable parameters.  In that regard, the paper is very appealing, as it shows that replacing linear layers with the butterfly networks does not result in any loss in performance.

The paper’s aim is to establish that linear layers can be replaced by butterfly networks and uses three different experiments to show this. The experiments do lend evidence to the claim that butterfly networks do not lead to a loss in performance.

The paper also present some theoretical results to show that the using butterfly networks sampled from the FJLT family preserves certain metric. However, my opinion is that Theorems 1 and 2 are mostly direct consequences of existing theorems in the sketching literature.

Finally, although I appreciate the experiments in the paper, I feel strongly that the paper lacks motivation. The idea of sketching and using FJLT transforms have had a great impact on numerical linear algebra and generally in reducing computational complexity of algorithms. However, it is unclear from this paper what the expected impact is. Probably the authors could focus more on this. Possible directions are to show improved training times, or maybe even showing that such networks are more resilient towards overfitting.

As interesting as the experiments are, the paper needs a better motivation and further exploration of the utility of the ideas presented. For example, can using butterfly networks improve theory or practice of deep learning/machine learning? I do like the paper, but the paper can be much stronger and much more appealing if the motivation is better justified.

---

> ### Author Response · Authors · 2020-11-17
> **Responses to the Queries Raised by the Reviewer**
>
> We appreciate the significant effort of all the reviewers to deliver such detailed and constructive suggestions. We also appreciate that most reviewers identify the novelty of our work. We have prepared a new version of the document by incorporating all the suggestions. Specific queries raised by Reviewer 1 are addressed below.
>
> 1) Theorems 1 and 2 are mostly direct consequences of existing theorems in sketching literature.
>
> Response: We note that our main theoretical contribution is Theorem 3 (renamed as Theorem 1 in the new version), which is on the optimization landscape of the encoder-decoder network where the encoder is replaced as mentioned in Section 5.  Also, as pointed out by Reviewer 2 we feel Theorem 3 (renamed as Theorem 1 in the new version) in our paper raises important questions about the optimization landscape of butterfly networks. As adequately noted by Reviewer 1, Theorem 1 (renamed as Proposition 1 in the new version to avoid confusion) can be proved easily using the existing theorems. We prove it for completeness in paper. Theorem 2 (renamed as Proposition 2 in the new version to avoid confusion) is from Sarlos[06]. We state and use it (without proving it) for motivating our proposed replacement of the encoder layer.
>
> 2) The paper strongly lacks motivation, and requires better motivation and further exploration of the utility of the ideas presented.
>
> Response: The motivations of this work are two-fold:
> a) practical aspect of deep learning: to reduce the number of weights from quadratic to (near) linear by replacing a dense linear layer in any neural network by the proposed butterfly architecture. This offers faster training and prediction in deployment while producing results that match and often outperform existing known architectures.
>
> In the new version of the paper we have added the training times (see Figures 10 and 11 in Appendix D.1) required in the experiments in Section 6.1 (as suggested by Reviewers 3 and 4). This aptly quantifies the improvements in training times in our experiments.
>
> b) theoretical aspect of deep learning: Since the proposed replacement adds logarithmic depth to the architecture, it might pose convergence related issues. Hence to study theoretically the problems posed by such a replacement. We take a small step towards this by studying the optimization landscape of an encoder-decoder network where the encoder is replaced as mentioned in Section 5. Also see response to point 1.
>
> To further demonstrate the utility of truncated butterfly networks, we consider a supervised learning approach as done by [IndykVY19], where we learn how to derive low rank approximations of a distribution of matrices by multiplying a pre-processing linear operator represented as a butterfly network, with weights trained using a sample of the distribution.
>
> We have used a part of the additional page to add these points in the ‘Discussion and Future Works’ section of the paper to better convey the motivation of our work. We have also added the point about resilience to over-fitting as a future direction.
>
> 3. It is unclear what the expected impact is.
>
> Response: We expect this paper will help both `theory and practice of deep learning’. As shown by the experiments of our paper the proposed replacement of a dense linear layer achieves the same loss and better training times compared to the original model.
> With respect to theory, the convergence question related to the proposed replacement cannot be handled by the current tools. We expect this would spurn more techniques to address convergence related questions in deep networks with butterfly networks. We have added a few lines stating the potential impact of our work in 'Our contribution' (renamed 'Our contribution and Potential Impact' in the new version)

---

### Decision · Program_Chairs · 2021-01-07
**Final Decision**

**Decision:**

Reject

**Comment:**

This paper shows that linear layers can be replaced by butterfly networks. Put simple, the paper follows the idea of sketching to design new architectures that can reduce the number of trainable parameters and also gives the theoretical and empirical analysis to validate this claim. In this regard, the paper would be  appealing.  But the theoretical results given in this paper are incremental.